# From Cybercrime to Digital Balance: How Human Development Shapes Digital Risk Cultures

**Răzvan Rughiniș** [1,2], **Emanuela Bran** [2,3,4,*], **Ana Rodica Stăiculescu** [1,3,5] and **Alexandru Radovici** [2]

1. Romanian Academy of Scientists, 010071 Bucharest, Romania; razvan.rughinis@upb.ro (R.R.); ana-rodica.staiculescu@fsas.unibuc.ro (A.R.S.)
2. Faculty of Automatic Control and Computers, National University for Science and Technology Politehnica Bucharest, 060042 Bucharest, Romania; alexandru.radovici@upb.ro
3. Doctoral School of Sociology, University of Bucharest, 010181 Bucharest, Romania
4. Neo Networking SRL, 031786 Bucharest, Romania
5. Faculty of Psychology, Behavioral and Legal Sciences, Andrei Saguna University of Constanta, 900196 Constanta, Romania
* Correspondence: emanuela.bran@s.unibuc.ro; Tel.: +40-724-021-149

**Abstract:** This article examines configurations of digital concerns within the European Union (EU27), a leading hub of innovation and policy development. The core objective is to uncover the social forces shaping technology acceptance and risk awareness, which are essential for fostering a resilient digital society in the EU. The study draws upon Bourdieu's concept of capital to discuss technological capital and digital habitus and Beck's risk society theory to frame the analysis of individual and national attitudes towards digital risks. Utilizing Eurobarometer data, the research operationalizes technological capital through proxy indicators of individual socioeconomic status and internet use, while country-level development indicators are used to predict aggregated national risk perception. Article contributions rely on individual- and country-level statistical analysis. Specifically, the study reveals that digital concerns are better predicted at a national level rather than individual level, being shaped by infrastructure, policy, and narrative rather than by personal technological capital. Key findings highlight a positive and a negative correlation between digital advancement with cybersecurity fears and digital literacy, respectively. HDI and DESI are relevant country-level predictors of public concerns, while CGI values are not. Using cluster analysis, we identify and interpret four digital risk cultures within the EU, each with varying foci and levels of concern, which correspond to economic, political, and cultural influences at the national level.

**Keywords:** cybersecurity; digital risk culture; risk society; technological capital; human development index; GCI; DESI

## 1. Introduction

As digitalization evolves, so do the experiences and the perceptions of digital hazards [1], leading to a multifaceted interaction involving consciousness, vulnerability, and remediation. With the growing integration of societies and individuals into the digital landscape, there is a corresponding rise in their vulnerability to cyber dangers such as security breaches and misinformation campaigns. At the same time, the increased prevalence of digital platforms has led to an enhanced recognition of these potential hazards, since individuals are better informed, and societies place a larger emphasis on the dissemination of knowledge on cybersecurity and general risk communication [2]. Concurrently, the progression of digitalization introduces increasingly sophisticated instruments and tactics to address these threats. Thus, this duality gives rise to a paradox: although heightened digital exposure has the potential to magnify perceived dangers, the augmented capacities for mitigation may diminish the impression of risk or even foster a state of complacency.

The ongoing evolution of technology creates an ambivalent and dynamic interplay between digitization and the perception of risk [3], leading to continuous changes and adjustments.

The higher the digitalization level (individually and collectively), the higher the exposure to more sophisticated risks, the higher the awareness of risks, and also, conversely, the higher the means to combat risks. This leads to an ambivalent relationship between the level of digitalization and the intensity of perceived digital risks. We explore this complex relationship with regards to several dimensions of perceived digital risks and safety, namely risks pertaining to privacy and cybersecurity, digital literacy and accessibility, child well-being and mental health, and environmental sustainability.

Our research questions explore the individual and collective forces that shape digital concerns:

- How does individual-level technological capital shape perceptions of digital risks?
- How does country-level human development (measured by the Human Development Index (HDI)) and digitalization (measured by the Digital Economy and Society Index (DESI) and the Global Cybersecurity Index (GCI)) shape cultures of digital risks?
- The article investigates the individual and collective dimensions in theory and empirically. The literature review focuses on the concept of technological capital, as derived from Bourdieu's discussion of capital and habitus, and on Beck's risk society. In order to model them for statistical analysis, we propose proxy variables as indicators for each of the two concepts. The methodology describes what type of analysis was performed, and the Section 4 presents the quantitative (numerical) and qualitative (visual) outcomes. The Section 5 explains the findings comparatively within the proposed conceptual framework, followed by a conclusion which highlights the main contributions of our paper.
- At the individual level, we find, in concordance with previous studies, that digital capital, as measured through socio-demographic proxies, does not strongly shape, on aggregate, public concerns of privacy and cybersecurity. This is largely due to the ambivalent nature of the relationship between capital and risk exposure and concerns, as detailed in the Section 1. Still, at the country level, we find significant differences. Study contributions consist of identifying HDI and DESI indices as relevant predictors for country-level variability in public concerns, especially for fears regarding cybersecurity. CGI values were not relevant predictors, possibly because of a data collection lag. We also contribute to the state of the art by identifying an exploratory typology of countries that we interpret as four digital risk cultures, each with a distinctive profile of concerns and with regional specificity.

## 2. Literature Review

### 2.1. Technological Capital and Digital Habitus

Concerns about digital technologies can have individual and a collective dynamics underlined by ambivalence. At an individual level, technological capital [4] could account for observed risk perceptions. This extension of Bourdieu's concept of capital refers to the resources that individuals hold, enabling them to engage with digital technology. The digital habitus of an individual would thus comprise the set of lasting dispositions based on personal experiences and assimilated perspectives that shape perceptions, appreciations, and action regarding the digital sphere [5]. The ease with which a person uses online tools, their digital consumption habits [6], and even their susceptibility or resistance to online threats are all influenced by their digital habitus. As societies become increasingly digital, a person's digital habitus interacts with their technological capital [7], affecting how they accumulate more of it and how they deploy it in different situations.

In light of Bourdieu's conceptualization of different social fields and forms of capital, we could talk about the four dimensions of technological capital [8]. The economic dimension is related to the availability of assets such as high-performance devices, premium membership subscriptions, and high-speed internet. The cultural dimension comprises skills and knowledge about the latest tech evolutions, including matters such as privacy

issues or having certifications in IT-related fields. Another dimension is represented by the social technological capital [9], which comprises membership in relevant networks and groups and having connections with influencers on social media and the tech industry. Finally, there is symbolic technological capital that captures the prestige of digital expertise and presence within the digital sphere.

Technological capital can also be classified into embodied, institutionalized, and material forms of capital. These refer to skills and competences that individuals control as embodied abilities, to their acquired degrees and certifications, and to their physical and digital assets, respectively. These forms of capital can be studied indirectly through proxies such as socio-economic variables and internet use when no direct indicators are available, such as in the case of the Eurobarometer dataset that we use.

The technological capital of individuals does not exist independently of their social position. It frequently interplays with their social, cultural, and economic capital, amplifying or attenuating the positives and negatives connected with each. For example, an individual with high economic capital can easily invest part of it in material technological capital, while an individual with strong social capital and limited technological capital might struggle to maintain their network of influential connections. Consequently, the conceptual relationship between technological capital and socio-economic variables is strong, which makes it possible to use the latter as proxy indicators for the former.

Frequency of internet use is a good measurement of embodied technological capital, as it implies familiarity and comfort with digital tools and services. This form of technological capital is also indirectly revealed in the age and the gender of an individual. The younger generations have been socialized as "digital natives", growing up immersed in the digital world and exhibiting intuitive interaction skills with it. Gender is also associated with STEM skills and digital savvy, though in variable forms and intensities, as boys and men are often more encouraged than girls and women to become acquainted with technologies and to invest in them emotionally.

Education indicates primarily a form of institutionalized technological capital, being an indicator of formal training and instruction in digital skills. Embodied technological capital could also be observed indirectly through education, social class, and community size, as they shape one's encounters with the latest digital advancements. Furthermore, along with the more direct proxy estimates of material technological capital through social class and community size, gender can also act as a proxy due to the existing economic inequalities.

### 2.2. Risk Society and Digital Development

Beck's "Risk Society" theory highlights the transforming nature of contemporary dangers and how they alter societal perceptions and priorities of digitalization [10]. The Human Development Index (HDI), the Global Cybersecurity Index (GCI) and the Digital Economy and Society Index (DESI) can be used as proxy indicators for the risk society. HDI captures socio-economic development, which is strongly linked to digital technology advancements [11], while DESI and GCI reveal society's digital engagement and cybersecurity preparedness, respectively. Analyzing how each component relates to the risk society formulates the basis for our second research question, which explores ambivalent relationships between digitalization and digital risks, at a collective level.

The first dimension of HDI is life expectancy at birth. This strongly relates to the medical infrastructure and other life sectors such as the food industry or work environment safety, all dependent on and enhanced by digital technologies. Mean years of schooling and expected years of schooling represent the second dimension of HDI. Higher levels of education can accommodate an advanced curriculum on technologies and their multifold impact on society [12]. The third dimension, GNI per capita, indicates economic prosperity. These financial resources are at the risk of being targeted through cybercrime, but they also provide means for developing increased security infrastructures.

The Global Cybersecurity Index (GCI) captures how well a society is equipped to withstand cybersecurity issues from five different dimensions. The first one assesses

the degree to which the legal system regulates data protection, critical infrastructures, and cybercrime. The next dimension focuses on national technical capabilities such as handling incidents by a Computer Incident Response Team (CIRT) and having Child Online Protection Reporting mechanisms. The third dimension watches for national cybersecurity strategies and agencies or organizations, with an additional oversight in online child protection. Another dimension measures capacity development such as conducting cyber-awareness initiatives, fostering R&D programs, and cultivating national cybersecurity industries. The last dimension assesses cooperation in the form of partnerships and bilateral or multilateral agreements between agencies, firms, and countries.

Moving forward to DESI, its first dimension is represented by Connectivity. Highly connected infrastructures introduce risks related to cyberattacks and the spread of misinformation. Conversely, digital coordination helps mitigate such issues. The second dimension is Human Capital focused on digital skills [13]. This simultaneously indicates a stronger reliance on digital technologies that could represent vulnerabilities, and a higher knowledge of secure digital practices that offer protection. Use of Internet Services by citizens represents the third DESI dimension. High internet engagement may create an increased access to knowledge and a higher digital footprint along with increased digital exposure, leading to cybersecurity and privacy issues [14]. The fourth dimension is Integration of Digital Technology by businesses. A digitalized private sector is more efficient and can accommodate new business models while being at the risk of data breaches and economic espionage. Digital Public Services is the fifth dimension of DESI, and higher digitalization poses a similar threat as the previous dimension. Digitalization brings risks such as data breaches and system failures.

### 2.3. Previous Studies on Public Perception of Privacy and Cybersecurity Issues

To our knowledge, at the date of manuscript submission (December 2023), there were no other studies that analyzed Eurobarometer 96.1 information concerning cybersecurity concerns, though Matefi [15] discusses Europeans' perceptions of their digital rights based on the same survey.

Still, a series of authors have analyzed other Eurobarometers and dedicated surveys concerning cybersecurity and privacy concerns. For example, Lee and Wang [16] analyzed Eurobarometer 2019 data on cybersecurity fears, identifying two types of Europeans based on individual levels of online activity and cybersecurity behavior (as reported in the survey): the "at-risk class" (with higher risk) and the "cautious class" (with lower risk). At the country level, they used as predictors the Global Cybersecurity Index (GCI), GDP per capita, internet penetration, and proportion of urban population, though only internet penetration was statistically significant in discriminating between the two groups, with higher rates leading to higher proportions of the "at-risk" type. The authors also find that, at the individual level, higher digital skills are, paradoxically, associated with the at-risk class, probably due to the ambivalent relationship mediated by exposure: "Surprisingly, changes in passwords, the maintenance of security settings, and concerns about cybersecurity have all been positively associated with risky Internet users. We speculate that members of the at-risk class might engage in more online activities, and while this would make them more predisposed to being targeted online, these individuals are likely also more self-aware and recognize the potential risks of their actions" (p. 22). In a different analysis of the same Eurobarometer 2019, Lee and Kim [17] conclude that fear of cybercrime is most strongly determined by individuals' prior victimization. This finding is also supported by a systematic review of fear of cybercrime conducted by Brands and Doorn [18]. This review also identifies gender as a correlate of cybercrime concerns, with women reporting higher subjective perceptions of risks, and a positive relationship between cybercrime concerns and protective measures online, conceptualized as "constrained behavior".

Zamfirescu et al. [19] have highlighted the ambivalent relationships between online activity, experiences of cybersecurity incidents, and concerns and preventive measures taken to address them, based on Eurobarometer 87.4/2017. Although the items analyzed are

different from Eurobarometer 2021 studied in this paper, and thus not directly comparable, the overall findings are consistent with our analysis. They classify European respondents into four attitudinal clusters, "avoiding", "engaging", "wary", and "aware". Similarly to our analysis, they show that socio-demographical differences as regards these types in relation to gender, age, difficulty of paying bills, and formal education are rather small. Still, countries differ markedly in the prevalence of the four types. This could possibly indicate the relevance of distinctive digital risk cultures that underlie individual attitudinal profiles. Lee and Kim [20] analyze similar data from a 2014 Eurobarometer and classify respondents into three types: uninformed users, disciplined users, and cautious users. They also conclude that country-level factors are better predictors of cybersecurity preparedness than sociodemographic factors, taking into account the GDP per capita and the Global Cybersecurity Index (GCI) values at the national level. Gomes and Dias [21] take a different approach to the Eurobarometer 2017 data, combining individual sociodemographic variables with internet use and the country-level Global Cybersecurity Index (GCI) into a multilevel factor model to predict an aggregated value of cybersecurity perceptions. They find that the GCI is a significant negative predictor for cybercrime risk perception, while individual-level predictors are significant just for self-confidence in one's abilities to use the internet and age (with negative associations) and buying goods online and male gender (with positive associations). We identify here a similar ambivalent connection, with higher digital capital indicated by GCI and self-confidence leading to lower concerns, while higher exposure indicated by online shopping and male gender leading to higher concerns.

In conclusion, previous studies regarding cybersecurity and privacy issues have highlighted the ambivalent relationship between technological capital and security concerns. Higher levels of capital enable effective action and protection, though they rely on more intense online exposure and experiences, which increase the risk surface. Starting from the review of the literature, we have chosen to add the DESI to our operationalization of the risk society, going beyond internet penetration measures studied before (which are also included in the DESI) and to examine all HDI dimensions, not just the GDP per capita, in an exploratory effort to find the best predictors that enable a modeling of cybersecurity concerns at the country level. Each dimension of the HDI, GCI, and DESI captures the ambivalent and mutual relationship between awareness level, mitigation capacity, and the outcomes of digital opportunities and threats characteristic of a risk society [22]. This ambivalent relationship between opportunities and risks was also highlighted by Bourdieu's theory of capital and its application to the digital field on an individual level. By exploring perceived digital threats through these two conceptual lenses, we will be able to trace how public opinion on digital risks is shaped.

## 3. Methodology

This paper is based on a secondary analysis of data collected through the Eurobarometer survey 96.1 from September to October 2021, part of which contains questions regarding digital rights and principles.

A central point of analysis throughout the paper is constituted by item QB3 "What worries you most about the increased role of digital tools and the internet in our society?", having several answer options: "Use of personal data and information by companies or public administrations", "Cyber-attacks and cybercrime such as theft or abuse of personal data, ransomware (malicious software) or phishing", "The difficulty of learning new digital skills in order to take an active part in society (e.g., working or studying online, online voting)", "The safety and well-being of children", "The difficulty some people have accessing the online world (e.g., persons with disabilities, elderly people, those living in areas with little or no internet access)", "The difficulty of disconnecting and finding a good online/offline life balance", "The environmental impact of digital products and services", "None of the above", "Other", and "Don't know". Each respondent could opt to choose or not each of these worries.

This article contains an individual-level analysis and a country-level analysis which were discussed comparatively in order to assess whether individual socio-economic status or the national properties of social structure and culture account more for the variations seen in digital worries. In the individual-level analysis, several variables were chosen as proxies for technological capital, including age, gender, age at graduation or present age if still studying, a dichotomous variable of whether somebody is currently a student or not, community size, social class, and internet use. For the country-level analysis, the HDI from 2021 and DESI from 2022 were chosen as proxies for the risk society.

The Section 4 includes the most relevant tables, while the Supplementary Material contains a more comprehensive presentation of findings. First, descriptive indicators of worries (QB3.1–QB3.7) and socio-demographic variables were obtained. Second, bivariate correlations were calculated for worries and socio-demographic variables, followed by correlations between these two categories of variables. Bivariate correlations between worries and socio-demographic variables were also performed within each of the EU27 countries. Third, two multinomial regression models were developed for predicting each of the seven worries, and their Nagelkerke Pseudo R-Square was registered. The first model (M1) includes all the previously mentioned socio-economic variables, while the second (M2) adds the country as a predictor.

The country-level analysis follows after the individual-level analysis. First, bivariate correlations were performed between each of the seven worries aggregated at country level and HDI along with DESI and GCI, both used as composite indices and separate dimensions. Second, the average mean of each worry for every country was calculated, on the basis of which an exploratory K-Means cluster analysis at the country level was carried out. Third, the four obtained clusters were interpreted according to their final cluster centers and visualized on a radar-style chart. Furthermore, the clustered countries were listed in a table and visualized on a geographical map. Cluster analysis is useful to overcome linear modelling by making possible ambivalent typologies that contain categories that vary along multiple dimensions [1,16,19,23–27].

## 4. Results

We proceed by first presenting descriptive statistics on the variables that represent worries captured in the Eurobarometer 96.1. For clarity, we listed them in Table 1 in descending order of the means. We observe that the order of perceived risk is issues related to cybersecurity, child safety, privacy, accessibility, life balance, digital literacy, and ecology.

Table 2 presents descriptive information for the socio-demographic variables used as proxy for individual measures of digital capital.

Next, in Table 3, we present bivariate correlations of the variables that represent worries and those that were chosen as a proxy for technological capital. In convergence with previous studies that found a low predictive value of individuals' socio-demographic position for their cybersecurity worries, such as Zamfirescu et al. [19] and Lee and Kim [20], we also find a low correlational relevance between these indicators. The analysis at an individual level displays rather low correlations, the highest values being 0.23 and 0.20. We observe that people who use the internet more have a slightly higher awareness of the dangers posed by cyber-attacks, use of personal data, and the difficulty of finding an online/offline life balance, on average, though differences highlighted by correlation coefficients are small. Furthermore, cyber-crime awareness increases with graduation age, and finding a balance is more of a concern for the younger population.

**Table 1.** Descriptive measures of items QB3.1–QB3.7. For all items, minimum value = 0, maximum value = 1.

| Variable | Mean |
|---|---|
| Cybersecurity: <br> QB3.2 Cyber-attacks and cybercrime such as theft or abuse of personal data, ransomware, or phishing | 0.56 |
| Child safety: <br> QB3.4 The safety and well-being of children | 0.53 |
| Privacy: <br> QB3.1 Use of personal data and information by companies or public administrations | 0.46 |
| Accessibility: <br> QB3.5 The difficulty some people have accessing the online world | 0.41 |
| Life balance: <br> QB3.6 The difficulty of disconnecting and finding a good online/offline life balance | 0.34 |
| Digital literacy: <br> QB3.3 The difficulty of learning new digital skills in order to take an active part in society | 0.26 |
| Ecology: <br> QB3.7 The environmental impact of digital products and services | 0.23 |

N = 26,521; cases have been weighted for a EU27 representative sample; minimum is 0; maximum is 1; because the variables are dichotomous, we do not report the standard deviation, which is redundant with the mean.

**Table 2.** Descriptive measures of socio-demographic variables.

| Variable | N | Minimum | Maximum | Mean | Std. Deviation |
|---|---|---|---|---|---|
| Age | 26,514 | 15 | 98 | 49.61 | 18.684 |
| Gender | 26,515 | 0 | 1 | 0.48 | 0.500 |
| Age at graduation or present age | 26,154 | 0 | 93 | 19.60 | 5.476 |
| Student dummy variable | 26,154 | 0 | 1 | 0.09 | 0.289 |
| Community | 26,516 | 1 | 3 | 1.98 | 0.752 |
| Social class | 26,019 | 1 | 5 | 2.49 | 0.976 |
| Internet use | 26,521 | 1 | 7 | 6.21 | 1.742 |
| Valid N (listwise) | 25,654 | | | | |

**Table 3.** Bivariate correlations between worries (QB3.1–QB3.7) and socio-demographic variables.

| Variables | Age | Gender | Graduation Age | Student Status | Community Size | Social Class | Internet Use |
|---|---|---|---|---|---|---|---|
| **Privacy** | **−0.10 \*\*\*** | 0.06 \*\*\* | **0.10 \*\*\*** | 0.02 \*\* | 0.02 \*\* | 0.05 \*\*\* | **0.17 \*\*\*** |
| **Cybersecurity** | **−0.12 \*\*\*** | 0.04 \*\*\* | **0.14 \*\*\*** | 0.06 \*\*\* | 0.01 | 0.09 \*\*\* | **0.23 \*\*\*** |
| **Digital literacy** | 0.05 \*\*\* | −0.02 \*\*\* | −0.05 \*\*\* | −0.04 \*\*\* | 0.00 | −0.04 \*\*\* | −0.01 \* |
| **Child safety** | −0.02 \*\*\* | −0.06 \*\*\* | −0.01 | −0.02 \*\*\* | −0.01 | −0.02 \*\*\* | 0.05 \*\*\* |
| **Accessibility** | 0.00 | −0.03 \*\*\* | 0.03 \*\*\* | −0.01 | 0.01 | 0.01 | 0.07 \*\*\* |
| **Life balance** | **−0.20 \*\*\*** | 0.02 \*\*\* | **0.10 \*\*\*** | **0.10 \*\*\*** | 0.05 \*\*\* | 0.09 \*\*\* | **0.16 \*\*\*** |
| **Ecology** | −0.08 \*\*\* | 0.00 | 0.05 \*\*\* | 0.05 \*\*\* | 0.05 \*\*\* | 0.07 \*\*\* | 0.08 \*\*\* |

N is between 26,019 and 26,521; statistically significant coefficients are marked with bold; $p \leq 0.05$ is marked \*; $p \leq 0.01$ is marked \*\*; $p \leq 0.001$ is marked \*\*\* Coefficients larger than 0.1 in absolute value are marked with bold.

The final step of the individual-level analysis is represented by multinomial analysis with two models, M1 and M2. Table 4 presents the obtained Pseudo R-Square Nagelkerke model values for the prediction of each digital worry. The low Nagelkerke values are not surprising, since correlations also produced rather low values. We observe that introducing the country as a predictor in the second model increases the Nagelkerke values for some of the predicted digital worries, reaching the highest value for concerns about cybersecurity.

The country captures cultural and infrastructural differences, accounting for variation in cybersecurity concerns that has been explained in previous studies through macro-level indicators such as the GCI or internet penetration, as documented by Lee and Wang [16], Lee and Kim [20], and Gomez and Dias [21].

**Table 4.** Multinomial regression models: summary of Pseudo R-Square Nagelkerke values.

| Models | Privacy | Cybersecurity | Digital Literacy | Child Safety | Accessibility | Life Balance | Ecology |
|---|---|---|---|---|---|---|---|
| Model 1: Socio-demographic variables + Internet use | 0.04 | 0.08 | 0.01 | 0.01 | 0.01 | 0.07 | 0.02 |
| Model 2: Socio-demographic variables + Internet use + Country | 0.06 | 0.12 | 0.04 | 0.04 | 0.04 | 0.08 | 0.05 |

Socio-demographic variables: age, gender, graduation age, student status, community size, and social class.

The country-level analysis displays more intense correlation values between worries and risk society proxies, rather than individual-level technological capital proxies. Within the section of Table 5 where HDI correlations are presented, we observe that aggregated country-level cybersecurity concerns strongly correlate with the income and education components. The digital literacy concern is more characteristic of lower-income countries. Furthermore, the accessibility concern is more often found in countries with higher-quality health systems, possibly a proxy for more solidaristic societies. Regarding the DESI correlations, the human capital, tech integration, and public service components strongly correlate with perceived cybersecurity risks and negatively correlate with digital literacy concerns. Concerns with the ecological impact are negatively correlated with the education component of HDI and with the public service component of DESI, which might indicate the higher trust and optimism in societies with higher levels of development on their society's capacity to handle the environmental impact of digital technologies. Still, most GCI dimensions and its aggregated value do not correlate with digital concerns at the country level. Only its legal and organizational measures components correlate positively with public concerns for digitalization's impact on child safety, possibly indicating the influence of public debates and controversies on this topic in countries with stronger legal and organizational policies for regulating the impact of digital technologies. It is also possible that the lower correlation values for GCI are derived from its two-year lag, since the latest available values are from 2020, while DESI values are available from 2022.

Figure 1 includes a visual representation of the pattern of associations between human development (HDI) and digitalization (DESI) and digital concerns, at the country level.

Figure 2 presents scatterplots for the highest correlation values at country levels, respectively, the HDI and DESI indices with public concerns with cybersecurity. The strong correlations derive from a linear relationship that can be noticed when country values are plotted against each other.

In the next step of our analysis, we performed an exploratory K-Means cluster analysis using the mean values of each worry at a country level. Previous typological analyses at the individual level highlighted two poles of high- and low-security exposure [16], with finer classifications capturing ambivalent intermediary types [19,20]. We opted for a four-class typology, which also included intermediary types of cultures of digital risks that combine high levels of concern on some dimensions with lower levels on other dimensions. Table 6 shows the obtained final clusters which we interpreted as distinctive digital risk cultures, using their specific profile of perceived risks and concerns. We proposed a name for each risk culture, taking into account its main focus. The table header contains the proportions of each configuration within the total EU27 population and the number of member countries, along with its distinctive characteristics. Figure 3 presents the obtained

values on a radar-style chart in order to better compare the four cultural profiles along the seven explored dimensions.

**Table 5.** Bivariate correlations between digital concerns (QB3.1–QB3.7) and HDI and DESI indices and components.

| Index/Digital Concerns: | Privacy | Cybersecurity | Digital Literacy | Child Safety | Accessibility | Life Balance | Ecology |
|---|---|---|---|---|---|---|---|
| HDI 2021—Total index | **0.46 *** | **0.77 **** | **−0.49 **** | 0.32 | **0.41 *** | **0.44 *** | −0.13 |
| HDI Health component | **0.49 **** | **0.52 **** | −0.21 | **0.38 *** | **0.60 **** | 0.38 | 0.09 |
| HDI Education component | 0.30 | **0.66 **** | **−0.48 *** | 0.11 | 0.12 | 0.34 | **−0.43 *** |
| HDI Income component | 0.30 | **0.73 **** | **−0.56 **** | 0.29 | 0.27 | 0.34 | 0.02 |
| DESI 2022—Total index | 0.31 | **0.83 **** | **−0.60 **** | 0.30 | 0.34 | 0.29 | −0.29 |
| DESI Human capital component | 0.29 | **0.84 **** | **−0.59 **** | 0.32 | 0.34 | 0.38 | −0.28 |
| DESI Connectivity component | 0.27 | **0.41 *** | −0.27 | 0.09 | 0.37 | 0.37 | 0.08 |
| DESI Tech integration component | 0.37 | **0.77 **** | **−0.42 *** | 0.23 | **0.50 **** | 0.31 | −0.26 |
| DESI Public service component | 0.16 | **0.70 **** | **−0.64 **** | 0.32 | 0.05 | 0.05 | **−0.39 *** |
| GCI 2020—Total index | −0.03 | 0.25 | −0.25 | 0.16 | 0.26 | 0.03 | −0.10 |
| GCI Legal measures | 0.11 | 0.36 | −0.37 | **0.41 *** | 0.27 | 0.11 | −0.13 |
| GCI Technical measures | −0.20 | 0.28 | −0.30 | 0.06 | 0.07 | −0.05 | −0.18 |
| GCI Organizational measures | 0.25 | 0.28 | −0.26 | **0.48 *** | 0.04 | 0.06 | 0.01 |
| GCI Capacity development | 0.05 | 0.16 | −0.06 | 0.06 | 0.33 | 0.22 | 0.05 |
| GCI Cooperative measures | −0.23 | −0.07 | −0.04 | −0.25 | 0.34 | −0.16 | −0.17 |

N is 27; statistically significant coefficients are marked with bold; $p \leq 0.05$ is marked *; $p \leq 0.01$ is marked **.

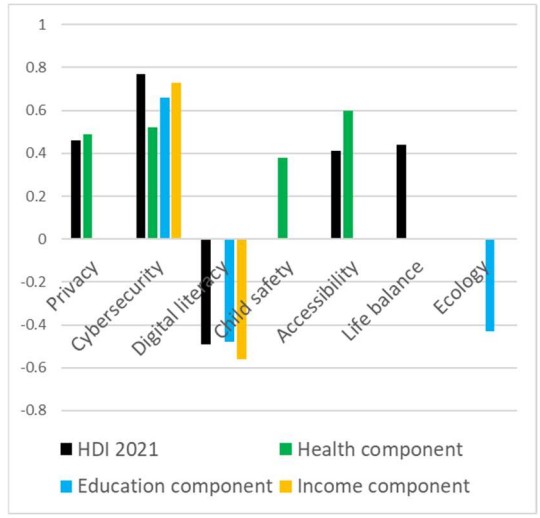
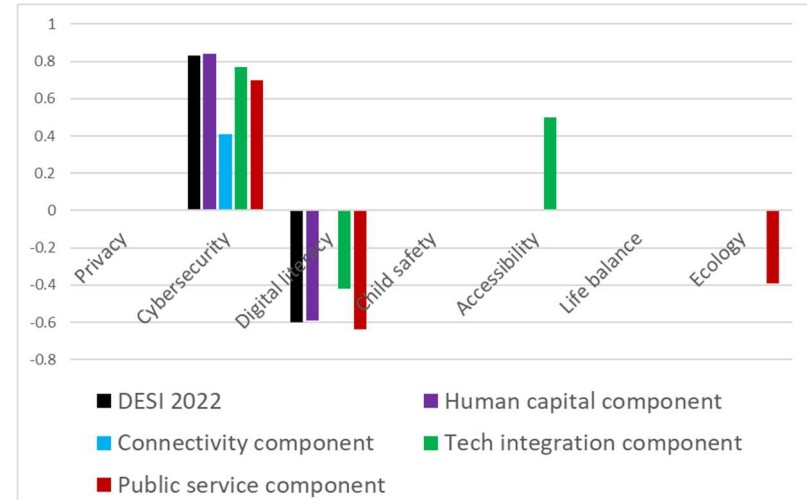

**Figure 1.** Visualization of correlation coefficients between digital concerns (QB3.1–QB3.7) and HDI and DESI indices and components (as presented in Table 5).

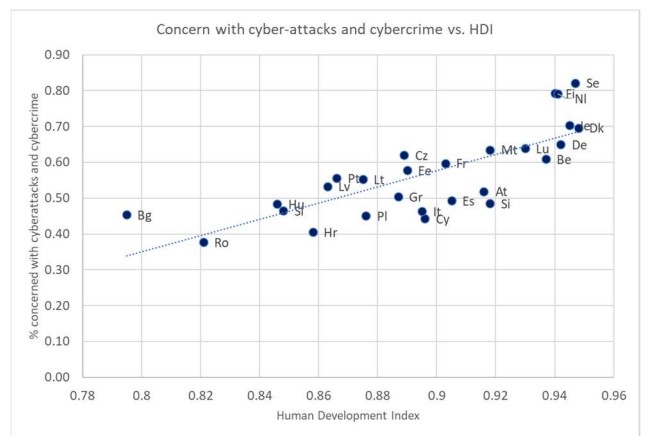
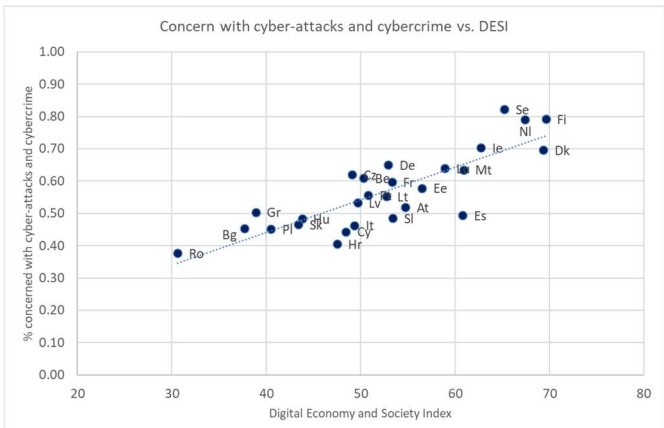

**Figure 2.** Scatterplot illustrating the correlations between public concerns with cyberattacks and cybercrime and HDI 2021 (Pearson correlation = 0.77) and DESI 2022 (Pearson correlation = 0.83) values, respectively, as presented in Table 5.

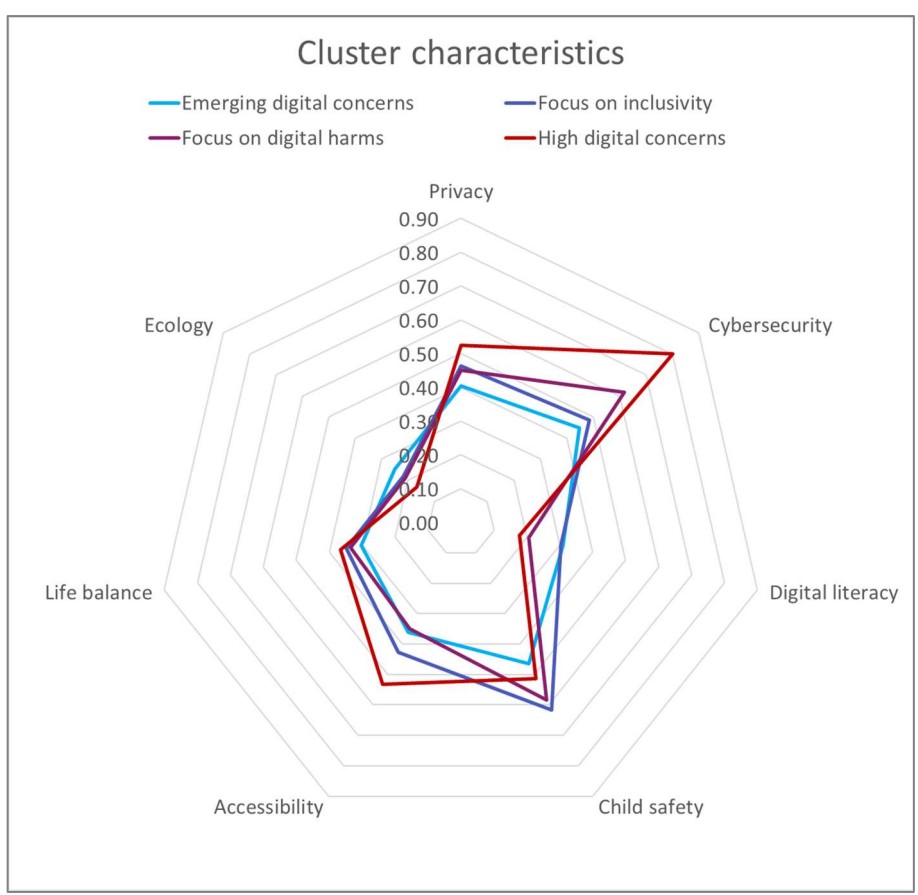

**Figure 3.** Radar-style chart of digital risk culture profiles.

**Table 6.** Cluster analysis results that highlight four digital risk culture profiles. The highest values per row are marked in bold.

| Variables | Digital Risk Culture 1: Emerging Digital Concerns 32% (6 Countries) | Digital Risk Culture 2: Focus on Inclusivity 19% (7 Countries) | Digital Risk Culture 3: Focus on Harms 42% (11 Countries) | Digital Risk Culture 4: High Digital Concerns 7% (3 Countries) |
|---|---|---|---|---|
| Privacy | 0.40 | 0.46 | 0.45 | **0.52** |
| Cybersecurity | 0.45 | 0.49 | **0.62** | **0.80** |
| Digital literacy | **0.31** | **0.30** | 0.21 | 0.18 |
| Child safety | 0.46 | **0.62** | **0.58** | 0.51 |
| Accessibility | 0.36 | **0.43** | 0.35 | **0.53** |
| Life balance | 0.30 | 0.35 | 0.34 | 0.37 |
| Ecology | 0.25 | 0.22 | 0.21 | 0.17 |

Header percentages represent proportions of each cluster from the EU27 population; the values represent final cluster centers; convergence was obtained on iteration 3. The largest values on each row, which define the specificity of the cluster, are marked with bold.

We notice that there are differences as well as commonalities between the four cultures of digital concerns. Cybersecurity ranks very high for all risk cultures, together with child safety and privacy, while ecological impact of digital technologies ranks lowest, with digital literacy and life balance having generally a low priority. Beyond these shared priorities, each culture has a distinctive focus. Countries with emerging digital concerns are specific in their relatively higher preoccupation with digital literacy, while countries with a focus on inclusivity prioritize child safety, accessibility, and digital literacy relatively more. Countries that focus on harm emphasize cybersecurity and child safety even more, while countries with high digital concerns are distinctive through their relative preoccupation with cybersecurity, accessibility, and privacy (see a synthesis in Table 7).

The following map shown in Figure 4 geographically delineates the identified digital risk cultures, and Table 7 contains the list of countries belonging to each configuration along with their highest concerns, listed in decreasing order of relevance within each cluster. The worries that are specific for each culture, by comparing them with the others, are marked with bold. Colors are only used to distinguish between cultural clusters, and numbering does not imply an ordinal type of variable.

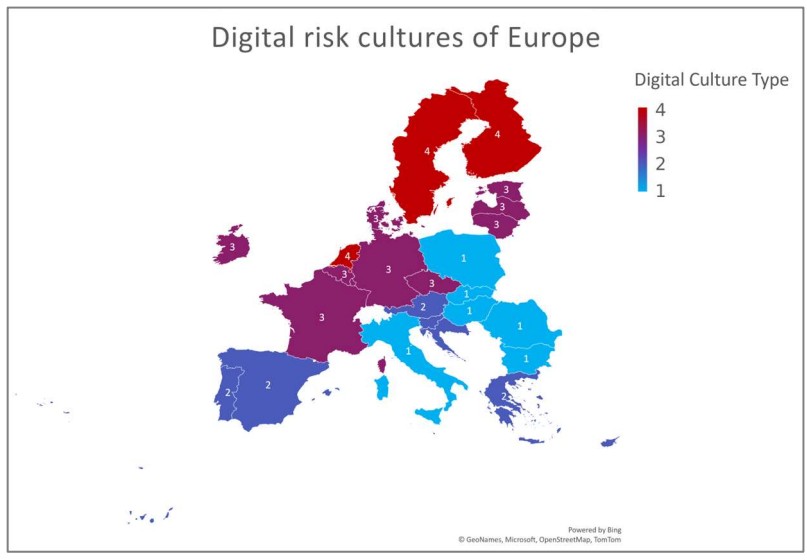

**Figure 4.** Geographical map of identified digital culture clusters.

**Table 7.** Countries belonging to each digital culture cluster.

| Culture | Digital Culture 1: Emerging Digital Concerns | Digital Culture 2: Focus on Inclusivity | Digital Culture 3: Focus on Harms | Digital Culture 4: High Digital Concerns |
|---|---|---|---|---|
| Digital concerns | Child safety<br>Cybersecurity<br>Privacy<br>Accessibility<br>**Digital literacy**<br>Life balance<br>Ecology | **Child safety**<br>Cybersecurity<br>Privacy<br>**Accessibility**<br>Life balance<br>**Digital literacy**<br>Ecology | **Cybersecurity**<br>**Child safety**<br>Privacy<br>Accessibility<br>Life balance<br>Digital literacy<br>Ecology | **Cybersecurity**<br>**Accessibility**<br>**Privacy**<br>Child safety<br>Life balance<br>Digital literacy<br>Ecology |
| Countries | Bulgaria<br>Hungary<br>Italy<br>Poland<br>Romania<br>Slovakia | Austria<br>Croatia<br>Cyprus<br>Greece<br>Portugal<br>Slovenia<br>Spain | Belgium<br>Czech Republic<br>Denmark<br>Estonia<br>France<br>Germany<br>Latvia<br>Lithuania<br>Luxembourg<br>Malta<br>Republic of Ireland | Finland<br>Sweden<br>The Netherlands |

Worries are listed in decreasing order by level of concern within the cluster; specific worries that are higher for each cluster in comparison with the other clusters are marked in bold; countries are listed in alphabetical order.

## 5. Discussion

The empirical data show that concerns over digital technologies emerge at a national and regional cultural level rather than as individual attitudes related to personal socio-economic status. These cultures of digital risks could stem from the complex interplay of the forces present in a risk society as well as elements such as the system of social norms and policies, values, and structures.

The risk society theory discusses the ambivalent connection between risk perception and social development. At a European level, we observe that the analyzed digital concerns are rather frequent, presenting a mean between 56% (around one in two people fear cybercrime) and 23% (around one in four fear an environmental impact of digital tech). This finding is in line with the EU being regarded as a highly developed digital arena, though with variations between different countries and social categories, where leading tech innovations and regulatory frameworks exhibit the dynamics characteristic of modern societies between risk production and management.

There are variations in the proportion of people that are worried about different digital risks at a European or country level. High concerns over cybersecurity, privacy, and child safety and wellbeing might reflect the perceived severity of negative outcomes. Other risks related to inclusivity, life balance, and the environment might be eclipsed in comparison regarding the perceived immediacy, personal experience, and broader social narratives of technological progress.

We observe that cybersecurity concerns strongly correlate with higher HDI and DESI and with their components, especially the income component of HDI and the human capital component of DESI. This further supports Beck's theory of risk production and perception in technologically advanced societies, as these people have more at stake to lose and also a higher awareness of consequences.

Privacy concerns seem to be related to a developed health sector, probably due to the sensitivity of such data and also to the type of society reflected by this indicator. To a lesser degree, this positive correlation is also observed in the tech integration component of DESI, as highly digitalized infrastructures also represent a possible system vulnerability. Privacy

concerns are lower when compared to cybersecurity, probably also because the EU is a leading policymaker in this area.

Accessibility shows a correlation pattern similar to privacy concerns but even stronger. Probably as health care standards rise, so too does the awareness towards digital inclusivity. And as integration of digital technologies becomes ubiquitous, so too does the fear of leaving behind certain individuals.

Digital literacy presents strong negative correlations with the HDI and DESI, especially with the economic component of HDI and the public service component of DESI. This might reflect the higher fear of the less digitally advanced populations of being left behind along with the trust of the digitally developed countries in the IT&C education opportunities.

Child safety, life balance, and ecology present overall lower correlation scores with country development indicators. Child safety is probably a universal concern not strongly related to development. Life balance concerns seem to arise more frequently in highly developed nations, while ecological concerns negatively correlate with educational development and digitalization of the public service. An explanation might be that the life pace in digitalized societies is sometimes disruptive to emotional health while these societies also better mitigate the ecological impact of such technologies.

The Eurobarometer data reveal four digital cultures that present distinctive profiles regarding perceived digital threats. The first digital culture we have identified extends within the geographical space of Central–Eastern Europe and Italy. The level of concerns within this culture is typically lower than the others, and the risk perception seems balanced across categories. This could be explained by these countries experiencing a transitional phase in their digital development. There is a slight emphasis on child safety and cyber-crime issues. The awareness of problems of privacy and accessibility proves the acknowledgement of challenges related to institutions and infrastructure.

The second digital culture is characteristic of Southern Europe and some Central European countries. It is characterized by a moderate overall concern profile with a focus on inclusivity. These countries place a higher emphasis on the safety and well-being of children along with a balanced online/offline life, possibly due to the cultural specifics. Furthermore, worries regarding accessibility for vulnerable populations stand out and suggest potential gaps in policy and infrastructure that need to be addressed. Medium-level concerns might suggest an awareness of the benefits of technological integration along with the acknowledgement of the pitfalls of possible disruptions.

The third digital culture occupies the space of Western Europe and includes the Baltic States and the Czech Republic. These countries also exhibit an intermediate level of digital concerns with an emphasis on possible harms of digitalization, as more advanced digital development creates awareness regarding the less anticipated risks. This deeper understanding of the complex influences of technologies is also reflected in the reduced concerns regarding digital literacy. The main characteristics of this region are a pronounced concern about cybersecurity issues followed by the safety of children online and lower concerns regarding digital skills. This is the mark of nations with advanced digital infrastructures and citizens with a consolidated digital education.

The fourth digital culture is spread across Northern Europe, where nations are some of the most digitally advanced in the world. Across these nations there is an extremely high level of concern about cyber-attacks and cybercrime. This can be explained by the advanced integration of digital services into daily life, which increases risk perception. Their second concern is accessibility, which is interesting given the actual level of development and indicates their high inclusivity standards. This region showcases Beck's idea of manufactured risks at its peak, having the highest exposure to sophisticated digital threats. While vigilant of the implications of cyberattacks, this cluster that represents the higher end of the spectrum of technological capital seems relatively comfortable with the evolution of the digital landscape related to environmental health.

Additional metrics that capture more dimensions of social development can enhance the understanding of variations in public attitudes towards cybersecurity and privacy

across nations. These dimensions could be derived from a comprehensive framework that incorporates economic, cultural, political, and social elements; each of these components may influence the way the public perceives and reacts to matters pertaining to cybersecurity and privacy. As an illustration, the economic stability of a nation could be quantified using a Socio-Economic Stability Index, which incorporates indicators such as employment rates, income inequality, and economic expansion. This is a fundamental component that could shape the way the general public perceives technology and security. Societies characterized by greater economic stability may allocate greater resources towards technological infrastructure and education, resulting in heightened consciousness and comprehension of cybersecurity vulnerabilities. On the contrary, in economies characterized by lower levels of stability, the emphasis on pressing economic concerns may eclipse the significance attributed to cybersecurity, thereby resulting in a diminished level of public attention towards these matters. The perception and response of populations to cybersecurity risks could be impacted by societal attitudes toward technology, which include trust in technological advancement and openness to adopting new technologies. This could be quantified using an index of Cultural Orientation towards Technology. Cultures that demonstrate a greater propensity to embrace technological advancements are more likely to adopt cybersecurity measures in a proactive manner. The public's perception of the government's transparency and level of trust, as assessed by a Government Transparency and Trust Index, could have a bearing on their attitudes towards cybersecurity initiatives lead by the state. A metric that quantifies the degree of community engagement and social cohesion could be a Social Cohesion and Community Engagement Score. Generalized trust is important in facilitating collaborative reactions to cybersecurity risks. Strong social cohesion increases the likelihood that members of a society will collaborate to defend against cyber threats and encourage communal protective behaviors. The degree to which a nation guarantees civil liberties and personal freedoms could be assessed by a Civil Liberties and Personal Freedoms Index; this, in turn, could influence public perceptions of privacy and cybersecurity. Societies that enjoy greater freedoms frequently prioritize personal privacy to the extent that they are more vigilant and responsive to cybersecurity threats that have the potential to violate these liberties. Conversely, civil liberties that are restricted may engender a diminished level of public opposition towards intrusive cybersecurity protocols. A Globalization and International Connectivity Index, measuring the degree to which a nation is integrated into global networks for trade, travel, and communication, could capture the correlations between globalization and vulnerability to and awareness of global cybersecurity issues. Countries with extensive connectivity are more prone to confronting a wide range of cybersecurity challenges; consequently, they may possess a more sophisticated comprehension and apprehension regarding these matters. Lastly, a factor of Historical Experience with Technology and Security Incidents would consider the ways in which a nation's prior encounters with security incidents associated with technology have influenced its present-day public sentiments and policies concerning cybersecurity. Nations with a history of substantial cybersecurity threats are more likely to possess a heightened level of awareness and more comprehensive policies regarding these risks. Conversely, countries without such a background may not perceive these threats with the same severity.

## 6. Conclusions

The article explores the configuration of digital concerns within the EU nations, a political and cultural space leading in innovation and digital policymaking. The importance of the research stands in discussing social forces that shape technology acceptance and risk awareness, two factors that sustain building a resilient society within the advanced digital landscape of the EU. The study investigates seven digital concerns related to privacy, cybersecurity, digital literacy, child safety and wellbeing, accessibility, online/offline life balance, and environmental impact.

The Section 1 is focused on two key concepts, namely Bourdieu's capital adapted to the technological field and Beck's risk society with its ambivalent links between risk

creation, mitigation, and perception. The Section 3 describes the Eurobarometer variables which are used as proxies for the technological capital within an individual-level analysis and those used as proxies for a risk society at a national aggregate level. Technological capital is operationalized through socio-economic variables and internet use that represent embodied, institutionalized, and material forms of capital. The indicators for the risk society that were explored comprise the HDI and DESI.

The Section 3 also presents the operations performed on the data including determining frequencies, computing bivariate correlations, exploring multinomial regression models, and classifying digital cultures through cluster analysis. The results revealed an accentuated variability in concerns at a national level rather than at an individual level. This implies that elements such as digital infrastructures, national policies, and broader narratives shape perceptions of the impact of technologies, rather than the digital habitus dependent on personal technological capital.

The main findings include a strong positive correlation between fear of cyberattacks and high digital development which supports Beck's theory regarding risk perception due to high exposure, deeper understanding, and irreversibility of impact. There is also a strong correlation between reduced digitalization and levels of digital literacy which further sustains the risk society theory and highlights the fear of being left behind. The cluster analysis at country level revealed four distinct digital cultures, each characterized by configurations of reduced concerns, moderate concerns with a focus on inclusivity, moderate concerns with a focus on harm, and high concerns. The specific digital risk awareness profiles align with four regions of the EU, revealing economic, political, and cultural forces that shape concerns at the national and regional levels.

One limitation of the research is with respect to the spread of the data, which is limited to the 27 countries of the EU. Even if culturally heterogeneous, the EU has a high cooperation between nations and a common strategy for technological innovation and good governance. Furthermore, studies are needed to analyze how these findings relate to the emergence of digital risk awareness within other nations around the globe. The second limitation of the study refers to the use of proxy variables for technological capital and risk society, as more direct measurements were not included in the Eurobarometer. A dedicated survey could better capture the distinctive social forces that shape public opinion on digital risks.

**Supplementary Materials:** R.R., E.B., A.R.S. and A.R. "From Cybercrime to Digital Balance: How Human Development Shapes Digital Risk Cultures. Supplementary Material". Available at https://bit.ly/3Gm8r3v (accessed on 16 November 2023).

**Author Contributions:** Conceptualization, R.R., E.B., A.R.S. and A.R.; methodology, R.R., E.B., A.R.S. and A.R.; software, E.B.; validation, R.R., E.B., A.R.S. and A.R.; formal analysis, R.R., E.B., A.R.S. and A.R.; investigation, R.R., E.B., A.R.S. and A.R.; resources, R.R.; data curation, E.B.; writing—original draft preparation, R.R., E.B., A.R.S. and A.R.; writing—review and editing, R.R., E.B., A.R.S. and A.R.; visualization, E.B.; supervision, R.R.; project administration, R.R.; funding acquisition, R.R. All authors have read and agreed to the published version of the manuscript.

**Funding:** This work was funded by the "Innovative Solution for Optimizing User Productivity through Multi-Modal Monitoring of Activity and Profiles—OPTIMIZE"/"Solutie Inovativa de Optimizare a Productivitatii Utilizatorilor prin Monitorizarea Multi-Modala a Activitatii si a Profilelor—OPTIMIZE" project, Contract number 366/390042/27.09.2021, MySMIS code: 121491, https://optimize.research-technology.ro/ (accessed on 16 November 2023).

**Institutional Review Board Statement:** Not required for secondary analysis of publicly available survey data.

**Informed Consent Statement:** Not required for secondary analysis of publicly available survey data.

**Data Availability Statement:** Publicly available datasets were analyzed in this study. (1) European Commission and European Parliament, Brussels (2022). Eurobarometer 96.1 (2021). GESIS, Cologne. ZA7846 Data file Version 1.0.0, Available at URL (accessed 1 September 2023): https://doi.org/10

.4232/1.13882. (2) UNDP (United Nations Development Programme), New York (2022). Human Development Report 2021-22: Uncertain Times, Unsettled Lives: Shaping our Future in a Transforming World, Available at URL (accessed on 1 September 2023): https://hdr.undp.org/informe-sobre-desarrollo-humano-2021-22. (3) European Commission, Brussels (2022). Digital Economy and Society Index (DESI) 2022. European Commission. Available at URL (accessed on 1 September 2023): https://digital-strategy.ec.europa.eu/en/library/digital-economy-and-society-index-desi-2022.

**Conflicts of Interest:** The authors declare no conflicts of interest. The funders had no role in the design of the study; in the collection, analyses, or interpretation of data; in the writing of the manuscript; or in the decision to publish the results.

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
