# Peer review of "From Cybercrime to Digital Balance: How Human Development Shapes Digital Risk Cultures"

_information, doi:10.3390/info15010050_

Round 1

Reviewer 1 Report

Comments and Suggestions for Authors

The article focuses on how social factors influence technology acceptance and risk awareness in the EU, a key area for innovation and policy. It uses Bourdieu's idea of capital and Beck's risk society theory to examine attitudes towards digital risks at both individual and national levels. The study employs Eurobarometer data and represents important findings. 

The literature review must be extended and the resluts must be compared to previous studies, highlighting historical data. Also, the selected metrics should be justified or supported with references. As of now, there is not enough clarification on the matter. 

Presenting the data from Table 4 in a graphical format would enhance its interpretability and visual appeal.

Another point is to support the statements with references, performing comparative analysis in the results section. 

Emphasize the need to refine or develop new metrics that could more accurately capture human development. The review should suggest specific metrics that could be relevant, along with a rationale for their suitability and potential impact.

Author Response

Thank you for the attentive reading of the manuscript and the constructive observations. We hope that we have addressed them, as presented in more detail below in "Answer to reviewer 1.docx".

Reviewer 2 Report

Comments and Suggestions for Authors

The article studies configurations of digital concerns within the European Union based on the Eurobarometer survey 96.1 from September-October, 2021 through statistical analysis. The study was performed to analyze this digital concerns in both individual and national levels. Following are reviewer's comments for the paper:

1. Authors requires that state contributions of the article explicitly in both abstract and first section of the article.

2. The descriptive measure shown in Table 1 requires more attention as the standard deviation are mostly big considering the scale between 0 and 1 used in the analysis. This result probably will have high impact on the correlation shown in Table 2. Authors requires to do more data extraction to reveal it's behaviour.

3. Most correlation values between the worries and socio-demographics variable in Table 2 are relatively small, this perhabs the direct consequence of high dispersion of the data. Even the highest resulting correlation 0.23 mentioned in the paper is considerably weak correlation. Therefore, it is a weak claim to say "We observe that the more someone uses the internet, the more they become aware of the dangers posed by cyber-attacks, use of personal data,...." in line 215 - 216. Authors might also need to show detail of the socio-demographics data used in this article. 

4. Authors requires to mention references as to where the four digital cultures terms used in the paper cited? On the other hand, the differences among all four digital cultures in terms of digital concerns shown in Table 6 are subtle. I wonder if the difference amount them are in terms of the order of the digital concerns. Authors should comment.

Author Response

Thank you for the attentive reading of the manuscript and the constructive observations. We hope that we have addressed them, as presented in more detail below in "Answer to reviewer 2.docx".

Round 2

Reviewer 2 Report

Comments and Suggestions for Authors

The authors have addressed the comments and fixed issues that were raised in the first version of the article. Now, the article should be suitable for publication.